# Elucidating Quadruplication Event of *PHO1* Gene: A Key Regulator of Plant Phosphate Translocation in *Brassica rapa*

**Dahlia Shahbuddin** [1,*] **, Rosazlina Rusly** [1] **, Ahmad Naqib Shuid** [2] **and Ahmad Bukhary Ahmad Khair** [1]

1   School of Biological Sciences, Universiti Sains Malaysia, Minden 11800, Penang, Malaysia
2   Advanced Medical & Dental Institute, Universiti Sains Malaysia, Kepala Batas 13200, Penang, Malaysia
*   Correspondence: dahliashah@usm.my; Tel.: +60-4-6534095

**Abstract:** In response to Pi deprivation, *phosphate 1* (*PHO1*) is a significant regulator at trans-eQTL hotspots in *Brassica rapa*. *Brassica rapa* short-read sequencing data analysis revealed four *PHO1* paralog genes, *PHO1_A*, *PHO1_B*, *PHO1_C*, and *PHO1_D*, placed in tandem with very high sequence similarity. However, based on short-read genomic sequence data, only three transcripts are accessible. Five bacterial artificial chromosomes (BACs) can be sequenced using a long-read sequencer, which improves de novo assembly and identifies structural variants. The *PHO1* gene's quadruplicating tandem positions in the genomic sequence were confirmed by an analysis of long-read data. Transcript analysis identified only three groups of *PHO1* paralogs (ortholog AT1G14040 in *Arabidopsis*), i.e., *PHO1_A*, *PHO1_B*, and *PHO1_D*, expressed in *B. rapa* leaf tissues under Pi deficiency. *PHO1_A*, with transcript ID XM_009150437.2, has five different splice variants found. These splice variants' truncated proteins demonstrated *PHO1_A*'s function in P control as opposed to protein encoding.

**Keywords:** *PHO1*; genetic regulator; phosphate deficiency; splice variants; *Brassica rapa*





## 1. Introduction

Phosphorus is not only a necessary component for all living things; it is also a crucial component of fertilizers that encourage plant development, making up 80% of the world's P consumption [1–3]. Plants only absorb P in the form of inorganic phosphate (Pi), which has a low concentration and a sluggish diffusion rate, from the rhizosphere solution [2,3]. When inorganic fertilizers are used excessively in agriculture, freshwater systems may eventually become eutrophicated and polluted [4–6]. Additionally, P is a finite resource that cannot be replaced in the cultivation of food crops. Mined rock phosphate is the main source of Pi fertilizer used in contemporary agriculture. However, the existing rock phosphate reserves could be depleted in 300–400 years based on current usage [7]. To alleviate these issues, P recovery and recycling are necessary to improve phosphorus use efficiency (PUE) in plants [1,8]. Therefore, plants have developed a range of metabolic and developmental adaptations to boost Pi acquisition in the soil [9,10].

During Pi deficiency, Pi is moved from mature leaves toward young, sink tissues within the plant [11]. It makes the plant require additional capacity to take up Pi from the rhizosphere. Therefore, multiple genes will be activated to promote Pi uptake and mobilization, including *Phosphate Starvation Response1* (*PHR1*), *Phosphate Transporter 1* (*PHT1*), and *Phosphate1* (*PHO1*) [12–14]. Phosphate uptake and transport are controlled by high-affinity phosphate transporters (PHTs). The PHTs are proton/phosphate (H$^+$, H$_2$PO$_4^-$) symporters, moving Pi ions through proton-transporter protein to traverse the cell membrane against the gradient. Proteins of the PHT family are grouped based on their sequence identity, cellular localization, and putative function into five classes (PHT1, PHT2, PHT3, PHT4, and PHT5). PHT1 subunits are the greatest high-affinity transporter and are located at the plasma membrane, and most of the *PHT1* genes show an increase in gene expression during Pi deficiency in plants [15,16].

The *Phosphate 1* (*PHO1*) gene family is involved in the Pi transportation from root to other plant organs through the vascular system of various tissues [13,17,18]. *PHO1*, which belongs to the SPX-EXS protein family, contains an SPX (SYG/Pho81/XPR1) tripartite domain in its N-terminal and an ERD1/XPR1/SYG1 domain in the C-terminal area important in maintaining Pi homeostasis [17,19]. In *Arabidopsis*, PHO1 localized to the endomembranes, mainly the Golgi, in root cells, mediating Pi loading into the xylem from root to leaf either by directly regulating the Pi transporter or through signal transduction [20]. The PHO1 family is the only protein that contains the SPX and EXS domains in eukaryotes. The major role of SPX is to modulate the activities of PHR1 to regulate *Phosphate Starvation Response* genes, while the EXS domain is important for exporting Pi. However, the EXS domain cannot function independently, at least in tobacco (*Nicotiana benthamiana*) [21]. In *Arabidopsis*, the *PHO1* gene family contains 11 members, namely, *PHO1* and *PHO1; H1-H10* [17,22]. Research on *Arabidopsis* revealed that *PHO1*, *PHO1; H1*, and *PHO1; H3* exhibit the same expression pattern, and all were expressed in the root cell and localized to the Golgi and trans-Golgi networks [22,23]. *B. rapa* has the largest *PHO1* gene family, represented by 23 genes, whereas soybean (*Glycine max*) has 14 putative *PHO1* homologs, and *Brachypodium* (*Brachypodium distachyon*) and maize (*Zea mays*) have only 2 *PHO1* homologs each [24].

In this study, we employ long-read sequencing to find the complex structural variants in the genomic regions of *Brassica* sp. by examining the potential genes at trans-eQTL hotspots. The long reads were processed to elucidate the four copies of *PHO1* genes located in tandem in the genome sequence. Furthermore, we defined these genes using transcript amplification and cloning to clarify the transcript presence. It is important to explain the genetic components responsible for phosphate deficiency to further manipulate the genes to improve phosphate use efficiency (PUE) in plants [25,26].

## 2. Materials and Methods

### 2.1. Identification of Bacterial Artificial Chromosome (BAC)

Five BAC clones, KBrB029J08, KBrB003E10, KBrB063F11, KBrH038K12, and KBrH102C10, from BAC libraries of *B. rapa* ssp. pekinensis cv. Chiifu, which contains the region of interest, were found using CloneFinder NCBI, "www.ncbi.nlm.nih.gov/clone (accessed on 20 July 2019)". The BACs were subcultured in 10 mL of Luria–Bertani (LB) media containing 12.5 µg mL$^{-1}$ chloramphenicol by shaking vigorously for 6–12 h at 37 °C.

### 2.2. Large-Construct BAC DNA Extraction

A single colony of BAC was inoculated in 10 mL LB with 12.5 µg mL$^{-1}$ chloramphenicol. Starter cultures were then shaken vigorously for 6–12 h at 37 °C. A 5 mL aliquot of the starter culture was added to 500 mL LB media and was mixed overnight at 37 °C for subsequent large-construct BAC DNA extraction. BAC DNA extraction was conducted using the Qiagen Large-Construct Kit (Qiagen, Manchester, UK) according to manufacturer's protocol. BAC DNA was then measured using Nanodrop 2000 spectrophotometer and Qubit fluorometer.

### 2.3. BAC DNA Precipitation

Each DNA sample received 0.1 volume of 3 M sodium acetate pH 5.2 (Thermo Scientific, Waltham, MA, USA) and 2.5 volumes of cold 100% ethanol. Then, the solution was vortexed and kept at −20 °C overnight to precipitate the DNA. DNA was pelleted using 12,000× *g* centrifuge at 4 °C for 20 min. The supernatant was removed, and then 500 µL of ethanol (70%) was added, inverted several times to wash the pellet, and vortexed briefly. This procedure was carried out twice. The DNA was pelleted in a 12,000× *g* centrifuge at 4 °C for 5 min. The pellet was briefly air-dried at room temperature after the supernatant was decanted. A total of 50 µL NFW was added to resuspend the DNA.

2.3.1. Preparation of Sequencing Library for MinION Sequencing

A 1 μg subsample of BAC DNA in 45 μL NFW was fragmented by loading the sample into a G-tube (Covaris, Brighton, UK). The sample was next centrifuged at $6000 \times g$ for 2 min before the tube was inverted and centrifuged again for 1 min. The fragmented DNA was repaired by adding 8.5 μL NFW, 6.5 μL FFPE repair buffer, and 2 μL FFPE repair mix (NEBNext FFPE RepairMix, NEB, Hitchin, UK). The corrected DNA was then cleaned up using 62 μL AMPureXP beads (Beckman Coulter, High Wycombe, UK) at room temperature and was eluted in 46 μL NFW. The cleaned and repaired DNA was quantified using a Qubit fluorometer with an expected recovery of greater than 1000 ng of material. SQK-LSK108 1D Ligation Sequencing Kit (Oxford Nanopore Technologies, Oxford, UK) with EXP-NBD103 Native Barcoding Kit (Oxford Nanopore Technologies) was used for the creation of ligation library (Table 1). The end-repair and dA tailing were performed using NEBNext Ultra II End-Repair Kit (NEB) in a total volume of 60 μL. The end-repaired DNA was cleaned up by incorporating 60 μL AMPure XP beads (NEB). The DNA was then quantified again using Qubit fluorometer with recovery target of 700 ng of material. The protocol for native barcoding genomic DNA was performed according to the manufacturer's instructions with few adjustments to guarantee high DNA recovery (Oxford Nanopore Technologies). Following the barcode ligation reaction, the DNA was cleaned again with AMPure XP beads and eluted in 26 μL of NFW. The amount of DNA added per BAC clone for library pooling was estimated based on 1 μL of DNA from the sample containing the lowest concentration. All other samples were added accordingly to produce an equimass pool of BAC DNA in the volume of 65.45 μL. A total of 1 μL of the sample was used for Qubit fluorometer measurement, and another 64.45 μL of this pooled DNA sample was used for adapter ligation using E6056 NEBNEXT Quick Ligation Module (NEB).

**Table 1.** Oxford Nanopore Technologies (ONT) barcodes are used for BAC sequencing.

| Barcode ID | Primer Sequence (5′-3′) | BAC ID |
|---|---|---|
| NB01 | AAGAAAGTTGTCGGTGTCTTTGTG | KBrB-029J08 |
| NB02 | TCGATTCCGTTTGTAGTCGTCTGT | KBrB-3E10 |
| NB03 | GAGTCTTGTGTCCCAGTTACCAGG | KBrB-63F11 |
| NB04 | TTCGGATTCTATCGTGTTTCCCTA | KBrH-38K12 |
| NB05 | CTTGTCCAGGGTTTGTGTAACCTT CCTA | KBrH-102C10 |

2.3.2. MinION Sequencing

The flow cell was inserted into the MinION device, and Platform QC was performed using MinKNOW software. The number of active pores was counted following the QC. A mixture of 504 μL running buffer with fluid mix buffer and 546 μL NFW was added into the sample loading port to prime the flow cell and was left for 5 min. A 14 μL aliquot of the modified and tethered library was filled into the sample loading port of the flow cell. Using the MinION control software MinKNOW (Oxford Nanopore Technologies), a 48 h sequencing protocol was initiated.

2.3.3. MinION Data Analyses

De novo assembly of individual BAC sequences was conducted using Canu program [27]. Three phases of data assembling were involved: 1. correction, where the quality of the bases in reads is checked and adjusted; 2. trimming, where the high-quality sequences were produced; and 3. assembly, where the reads are used to generate contigs and consensus sequences. This software facilitates nanopore sequencing by improving the quality of sequence assembly [27]. The sequences were then aligned to look at the plasmid backbone and the BAC ends using CLC Sequence Viewer software 7.8.1 (Qiagen). Then, the plasmid sequence was eliminated to obtain the true size of the BAC DNA sequenced. The size of each BAC was determined and compared to its predicted size. The target genes (*PHO1*) were aligned to each BAC using CLC Sequence Viewer software 7.8.1 (Qiagen).

### 2.4. Identification and Characterization of PHO1 Homolog Genes

Hammond et al. 2011 [25] revealed candidate genes at trans-eQTL hotspots on chromosome A06 *B. rapa*. Analysis showed the *B. rapa* genome contained four copies of *PHO1*, a known regulator of plant responses to Pi availability [22,28]. However, only three copies are evident from short-read transcriptome data (Figure 1). The efficiency of short-read sequencing for annotation of extremely similar copies is needed to identify the transcript's existence. Therefore, *PHO1* transcripts were amplified and cloned.

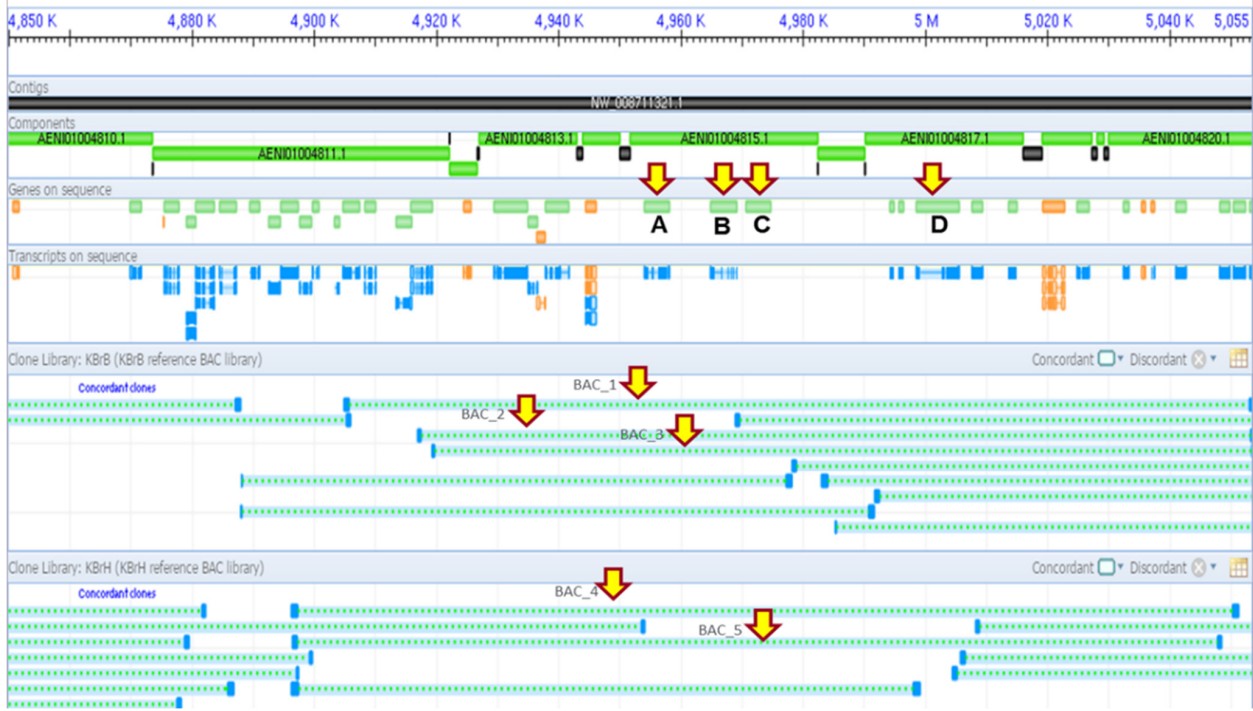

**Figure 1.** Four candidate gene locations at the trans-eQTL hotspots on Chromosome A06 of *Brassica rapa* and five BAC clones covering a region of interest from the *Brassica rapa* clone finder "www.ncbi.nlm.nih.gov/clone/ (accessed on 15 July 2021)". All four genes were, namely, A = *PHO1_A* (*Bra019686*), B = *PHO1_B* (*Bra019688*), C = *PHO1_C* (*Bra019689*), and D = *PHO1_D* (*Bra019690*).

Alignment of nucleotide (nt) sequences of *B. rapa PHO1* homologs XM_009150437 (*PHO1_A*), XM_018652610 (*PHO1_B*), XM_018652651 (*PHO1_C*), and XM_009150438 (*PHO1_D*) revealed three conserved regions which required an effective primer design. A 21 nt long left primer containing two degenerate bases was designed incorporating the start codon, and two right primers were designed to capture > 2 Kb of *PHO1* transcript(s), one of which is designed from a complete consensus region among the four *PHO1* mRNAs.

Nucleotide sequences of *B. rapa PHO1* homologs XM_009150437, XM_018652610, XM_018652651, and XM_009150438 were obtained from the GenBank database (NCBI). The CLC Sequence Viewer (Qiagen) was used to align the sequences. Two sets of primers were built from conserved regions in the alignment to harbor the coding sequence of the four *PHO1* homologs (Table 2).

**Table 2.** A. List of primers for cDNA amplification. B. The *PHO1* cDNA expected sizes.

| A. Primer | Sequence (5′ to 3′) |
|---|---|
| NB01 | AAGAAAGTTGTCGGTGTCTTTGTG |
| NB02 | TCGATTCCGTTTGTAGTCGTCTGT |
| NB03 | GAGTCTTGTGTCCCAGTTACCAGG |
| NB04 | TTCGGATTCTATCGTGTTTCCCTA |
| NB05 | CTTGTCCAGGGTTTGTGTAACCTT CCTA |

**Table 2.** *Cont.*

| B. Target genes | Expected size of cDNA (bp) |
| --- | --- |
| *PHO1_A* | 2224 |
| *PHO1_B* | 2044 |
| *PHO1_C* | 2152 |
| *PHO1_D* | 2227 |

### 2.4.1. Reverse Transcription Polymerase Chain Reaction (RT-PCR)

Amplification was carried out in a 20 μL reaction containing 10 μL of PCR Phusion Flash high-fidelity master mix (Thermo Scientific, Swindon, UK), 2 μL of 5 μM forward primer, 2 μL of 5 μM reverse primer, 3 μL of first-strand cDNA (10× dilution), and 3 μL NFW. The following gradient cycle was used for PCR amplification: 1 cycle at 98 °C for 1 min; 35 cycles at 98 °C for 1 s, 65 °C for 5 s, and 72 °C for 45 s; and 1 cycle at 72 °C for 1 min. Electrophoresis gel was performed using 1.2% agarose TAE gel along with Fast Ruler Middle Range DNA Ladder (Thermo Scientific) for 2 h at 80 V after PCR amplification.

### 2.4.2. Cloning of PHO1 Transcripts

The *PHO1* transcripts were combined and cloned into pCR4 Blunt TOPO vector using ZeroBlunt TOPO Kit (Invitrogen, Waltham, MA, USA), following the manufacturer's instructions. In a 0.2 mL PCR tube, a total of 1 μL (24 ng) of purified PCR product was added, combined with 1 μL of vector, 1 μL of salt solution (1.2 M NaCl and 0.06 M MgCl$_2$), and 3 μL of NFW for ligation stage. The tube was gently mixed before being centrifuged for 2 min. The tube was then incubated at 22 °C for 30 min before it was immediately chilled on ice. The transformation stage began with the addition of 2 μL of ligation mixture to a single tube of chemically competent Top10 cells (ThermoFisher, Horsham, UK). The tube was kept on ice for 30 min. The transition took place in a dry bath with a heat shock of 42 °C lasting 50 s. The tube was chilled on ice for 2 min before adding 250 μL of Super Optimal Catabolite medium (Sigma-Aldrich, Singapore) to improve plasmid transformation efficiency. The culture was incubated at 37 °C for an hour while being shaken horizontally at 225 rpm on an orbital shaker. Before being incubated overnight at 37 °C, 20 and 200 μL of the culture were spread on two LB agar plates containing 50 mgL$^{-1}$ of kanamycin. A total of 30 colonies were selected and subsequently grown in LB liquid supplemented with 50 mgL$^{-1}$ kanamycin for 14–16 h at 37 °C with low shaking at 250 rpm. The isolation of plasmids was conducted using GeneJET Plasmid Kit (ThermoFisher, UK).

### 2.4.3. Restriction Analysis

The plasmids were evaluated using HindIII restriction enzyme. For restriction, 1 μg of plasmid DNA was digested in a 20 μL reaction containing 1 μL of enzyme and 2 μL of 10× restriction buffer. The restriction reaction was mixed and incubated at 37 °C for 1 h. Digested plasmids were run on 1.2% agarose TAE gel at 80 V for 1 h.

### 2.4.4. Plasmid Sequencing

The plasmids were chosen based on the restriction fingerprints. Then, the selected plasmids were sent for sequencing using Sanger sequencing method to Source Bioscience (Nottingham, UK). The sequencing results were examined using SeqMan (DNASTAR, Madison, NY, USA).

### 2.4.5. Phylogenetic Tree Construction

All sequences were aligned, and the neighbor-joining method was used to perform cluster analysis. The phylogenetic tree was created based on the distance data using CLC Sequence Viewer 7.8.1 (Qiagen) software.

## 3. Results

### 3.1. PHO1 at Trans e-QTL Hotspots on Chromosome A06 B. rapa

The eQTL of Pi starvation responses was identified in the *B. rapa* mapping population (IMB 211 × yellow sarson R500). The result showed that the number of eQTLs was larger than anticipated according to physical and genetic maps on chromosomes A06 and A01, where Gene Ontology terms were high with P metabolism-related genes in A06 and high with chloroplast and photosynthesis-related terms in A01. A list of genes significantly differentially expressed in response to Pi deficiency was generated based on microarray data of Pi-deficient B. rapa R-o-18. Microsoft Access was used to compare this microarray list of genes to a list of genes in the genomic region surrounding the eQTL hotspot on chromosome A06. This comparison reveals the genes whose expression was influenced by alteration of Pi availability in this region. Four candidate genes were discovered to be the main regulatory trans-eQTL hotspot and located in tandem. All these genes belong to the PHO1 gene family. However, short-read transcriptome data revealed only three copies of genes appeared (Figure 1). Therefore, five BAC clones spanning the region of interest, KBrB029J08, KBrB003E10, KBrB063F11, KBrH038K12, and KBrH102C10, were identified and sequenced using long-read MinION sequencer from Oxford Nanopore Technologies to elucidate the locality of the genes. Moreover, to gain complete insight into *B. rapa PHO1* gene, the transcripts were amplified and cloned.

### 3.2. Candidate Genes Underlying Trans-eQTL Hotspots

Sequence data analysis revealed the four candidate genes were *Bra019686* (*PHO1_A*), *Bra019688* (*PHO1_B*), *Bra019689* (*PHO1_C*), and *Bra019690* (*PHO1_D*). The alignment was conducted using Geneious software, and the result showed high similarities among all four candidate genes (Figure 2).

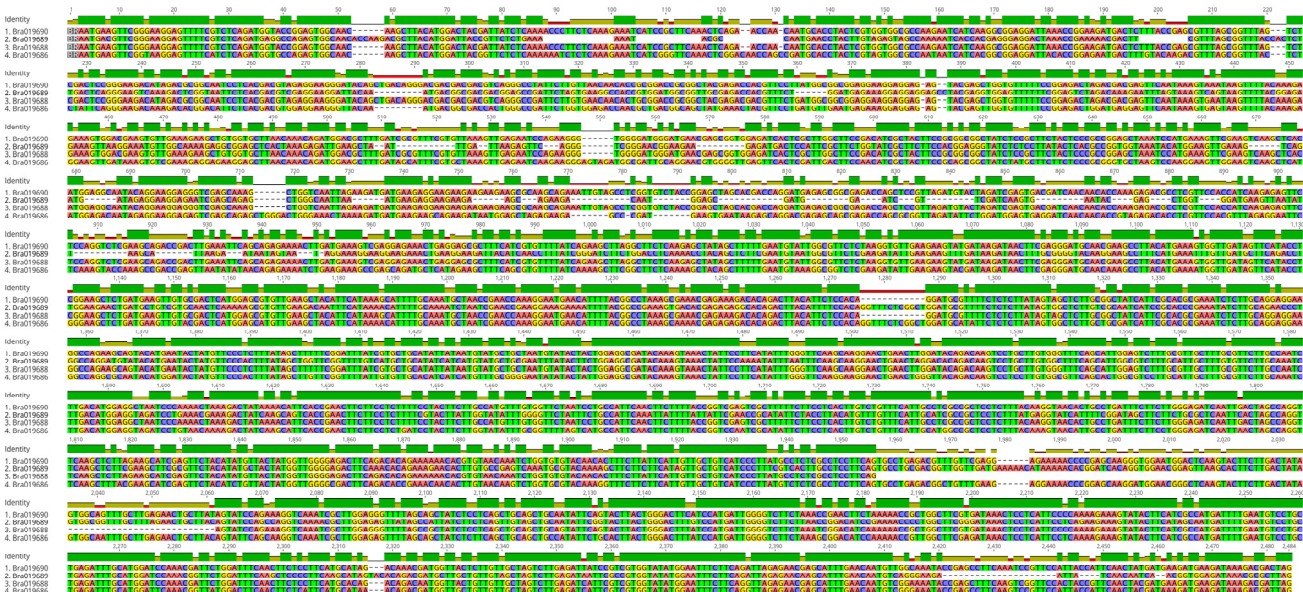

**Figure 2.** Sequence alignment of four putative *PHO1* sequences in *Brassica rapa* extracted from Brassicaceae Database (BRAD) "http://brassicadb.cn (accessed on 17 May 2022 )". to determine the region of homology (Geneious software 2022.1). The high similarity of the sequences is shown in green; yellow is for less similarity, and red refers to very low similarity. *Brassica rapa* PHO1 ID; 1. *Bra019690*, 2. *Bra019689*, 3. *Bra019688*, and 4. *Bra019686*.

*PHO1_A*, *PHO1_B*, *PHO1_C*, and *PHO1_D* are the four possible transcripts in this region of the genome. From the database, there is no transcript evidence of *PHO1_C*. Five BAC clones spanning the region of interest, named KBrB-063F11 (BAC 1), KBrH102C10 (BAC 2), KBrB029J08 (BAC 3), KBrH038K12 (BAC 4), and KBrB003E10 (BAC 5), were suc-

cessfully sequenced using Nanopore sequencer. The size of BACs ranged from 134,576 bp (BAC 1) to 155,059 bp (BAC 4) following sequencing.

### 3.3. BAC DNA Sequencing Using MinION Sequencer from Oxford Nanopore Technologies

A total of 991,846 reads were generated, resulting in a total of 6.4 Gb of sequenced bases. The number of reads generated from quality sequencing summary was 860,433, with about 5.7 Gb of total bases sequenced. The actual BAC sequence was achieved after the removal of the plasmid sequence and aligned the plasmid backbone with BAC ends using CLC software.

Comparison of the projected size of BAC (from NCBI clone finder) with the actual size from MinION sequencing determined some variations (Table 3). BAC 1 (KBrB-63F11) showed an increase in size, from 134,576 bp to 184,651 bp. In BAC 2 (KBrH-102C10), the real size from MinION sequencing was revealed to be marginally greater (1 Kb) than the expected size of 152,113 bp. BAC 3 (KBrB-029J08) and BAC 4 (KBrH-38K12) showed an increase in actual size to 185,935 bp and 165,146 bp, respectively. BAC 5 (KBrB-3E10) showed a shorter BAC length compared to the predicted size, recording only 45,485 bp. This might be attributed to the MinION device stopping during the 48 h run. Consequently, the incomplete sequence of BAC 5 was omitted from future research. Interestingly, from the five BACs sequenced and analyzed, there was only a consensus found in BAC 1 (KBrB-63F11) and BAC 3 (KBrB-029J08). Both BACs are from the same library, KBrB, and share identical BAC ends. Therefore, these BACs were chosen for further studies. The findings revealed that all four target genes (PHO1_A, PHO1_B, PHO1_C, and PHO1_D) were present and arranged in tandem after aligning all four target genes to each BAC (Figure 3).

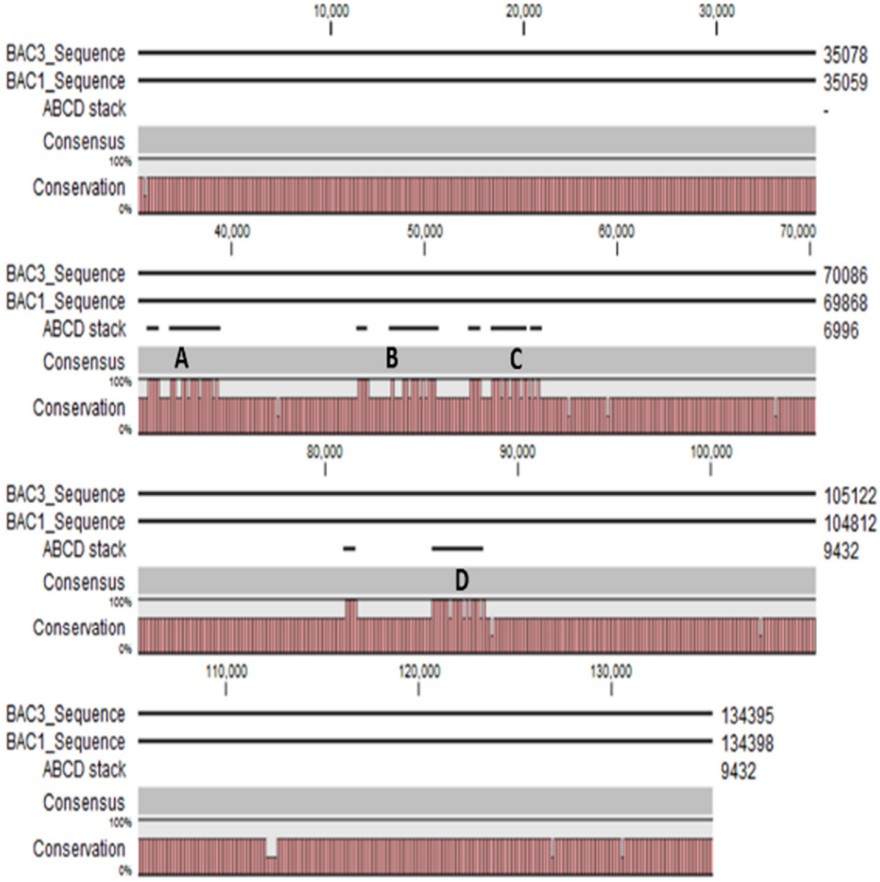

**Figure 3.** Locations of *PHO1* homolog genes mapped on BAC 3 and BAC 1 sequences viewed using CLC Sequence Viewer 7.8.1 (Qiagen). A. *PHO1_A*, B. *PHO1_B*, C. *PHO1_C*, and D. *PHO1_D* on BAC 1 and BAC 3 sequence.

**Table 3.** BAC size comparison of predicted and MinION sequenced BAC.

| ID | BAC | Predicted BAC Size (bp) | BAC Length (bp) |
|---|---|---|---|
| 1 | KBrB-63F11 | 134,576 | 184,651 |
| 2 | KBrH-102C10 | 152,113 | 153,335 |
| 3 | KBrB-029J08 | 148,972 | 185,935 |
| 4 | KBrH-38K12 | 155,059 | 165,146 |
| 5 | KBrB-3E10 | 136,929 | 45,485 |

### 3.4. Identification and Characterization of PHO1 Homolog Transcripts

A gradient PCR was conducted at 60 °C and 65 °C annealing temperature using cDNA from P+ and P− treatments. Both primer sets amplified several bands in anticipated size ranges at 60 °C (Figure 4). However, primer sets PHO_1_F and PHO_2224_R produced brighter bands (Figure 4A). A band of unexpected size of around 4 Kb was also amplified, which might have resulted from alternative splicing of *PHO1* or could be an entirely nonrelated transcript.

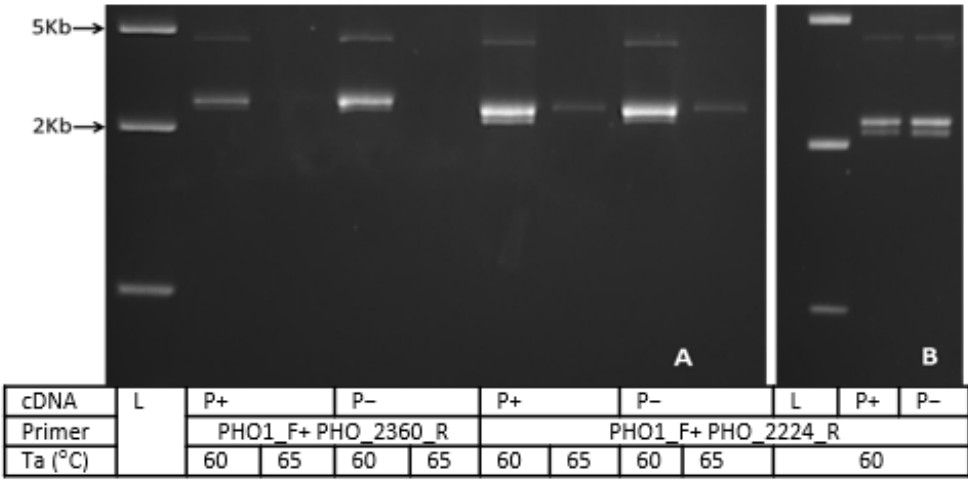

**Figure 4.** Electrophoresis gel image of PHO1 Polymerase Chain Reaction (PCR) of cDNA from *Brassica rapa* R-o-18 leaf samples. (**A**). Lane 1 from left: FastRuler Middle Range DNA Ladder (L); lanes 2 to 9 is PCR for cloning *PHO1* transcripts. (**B**). Gel of an aliquot of the PCR products after optimizing PCR.

### 3.5. Colony PCR and Restriction Analysis

Thirty colonies were analyzed using colony PCR, and the gel image of the colony PCR products revealed several size variants (Figure 5). For precise size determination, colony PCR products were subjected to HindIII restriction enzyme digestion to distinguish between the four *PHO1* homologs. The colony PCR restriction analysis verified the size variants found in colony PCR (Figure 5). The variant type 1 (colony 1–6, 8–11, 13, 15, 17–21, and 25–30) was prevalent among the colonies examined. Other variants are type 2 (7 and 14), 12, 16, 22, 23, and 24. Selected colonies were cultured overnight, and plasmid DNA was isolated and subjected to restriction confirmation and subsequent Sanger sequencing.

The sequencing results of seven plasmids showed a high-quality single read. BLASTN study (with *B. rapa* cultivar Chiifu 401–402) showed that transcript 2 was 96% identical to *B. rapa* phosphate transporter *PHO1* homolog 3 (XM_009150437). There were 94 SNPs and 2 indels that were different between the two sequences. The transcript 2 sequence was 2236 bp long and predicted to encode a 744 amino acid peptide. BLASTP analysis revealed 98% similarity between the protein encoded by transcript 2 and *B. rapa* phosphate transporter *PHO1* homolog 3 isoform X1 (XP_009148685) (Table 4).

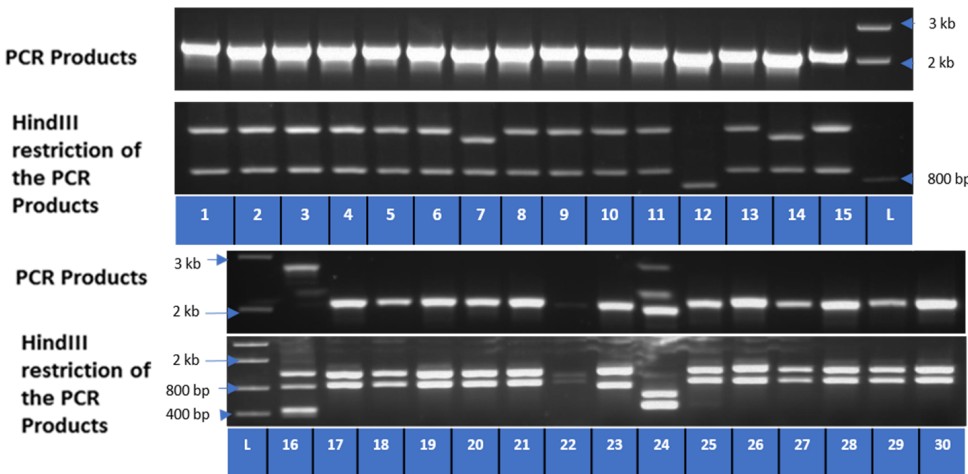

**Figure 5.** Gel image of colony PCR and HindIII restriction analysis of colony PCR of 30 randomly selected colonies containing PHO1 pCR4 BLUNT TOPO vector. Bands < 80 bp are not shown.

**Table 4.** Plasmid sequenced with the transcript number, nucleotide length, peptide length, closest BLASN and BLASTP hits, and predicted *PHO1* on chromosome A06 *Brassica rapa*.

| Transcript No. (NCBI) | Plasmid | Nucleotide Length | Peptide Length | Closest BLASTN Hits | Closest BLASTP Hits | Predicted *PHO1* |
|---|---|---|---|---|---|---|
| XM_009150437.2 | 2 | 2236 | 744 | XM_009150437.2 | XP_009148685.1 | A |
| XM_009150437.2 | 7 | 2150 | 552 | XM_009150437.2 | XP_018508167.1 | A |
| XM_009150437.2 | 23 | 2130 | 629 | XM_009150437.2 | XP_009148685.1 | A |
| XM_009150437.2 | 26 | 2317 | 746 | XM_009150437.2 | XP_009148685.1 | A |
| XM_009150437.2 | 22 | 2073 | 490 | XM_009150437.2 | XP_018508167.1 | A |
| XM_018652610.2 | 12 | 2092 | 697 | XM_018652610 | XP_018508125 | B |
| XM_009150438.3 | 24 | 2042 | 516 | XM_009150438 | XP_009148686 | D |
| XM_009150438.3 | 16 | 513 | 170 | XM_009150438 | - | |

BLASTN search for 2092 nucleotide long sequence of transcript 12 (using *B. napus*, cultivar ZS11 nucleotide sequence) revealed 99% similarity with *B. napus* phosphate transporter *PHO1* homolog 3-like mRNA (XM_013785900), which differed by only 17 SNPs. The closest hit in *B. rapa* was phosphate transporter *PHO1* homolog 3 transcript variant X3 mRNA (XM_018652610.1), which had 80% coverage and 90% homology. The transcript 12 sequence may encode a 697 amino acid (aa) peptide that shares very high conservation (99%) with *B. napus* phosphate transporter *PHO1* homolog 3-like mRNA (XP_013641354). Surprisingly, *B. rapa* had only 66% homology with phosphate transporter *PHO1* homolog 3 isoform X2 (XP_018508125).

The length of transcript 24 was determined to be 2042 bases. BLASTN identified a 97% similarity with *B. rapa PHO1* homolog 3-like mRNA (XM_009150438). However, this sequence only encodes a 516 aa protein, whereas XM_009150438 encodes an 811 aa protein (XP_009148686). The first 415 amino acids of both proteins were found to have 97% similarity in multiple alignments. Exon-to-exon alignment revealed that part of the exon 7 of XM_009150438 is missing in transcript 7, suggesting that alternative splicing may have caused a truncated protein. However, because the DNA and protein sequence of XM_009150438 and transcript 24 are substantially similar, it was established that transcript 24 is *PHO1_D* homolog.

Multiple alignments of seven chosen transcript sequences revealed transcript 2, which proved to be *PHO1_A,* which has four more splice variants, including transcripts 7, 22, 23, and 26, each of which is 2150, 2073, 2130, and 2317 bp long; their alignment is shown in Figure 6. Transcripts 7, 22, and 23 encode truncated proteins of 552, 490, and 629 aa, respectively, as a result of alternative splicing. Transcript 26 was 2317 bp long and might encode a protein of 746 aa (Table 4).

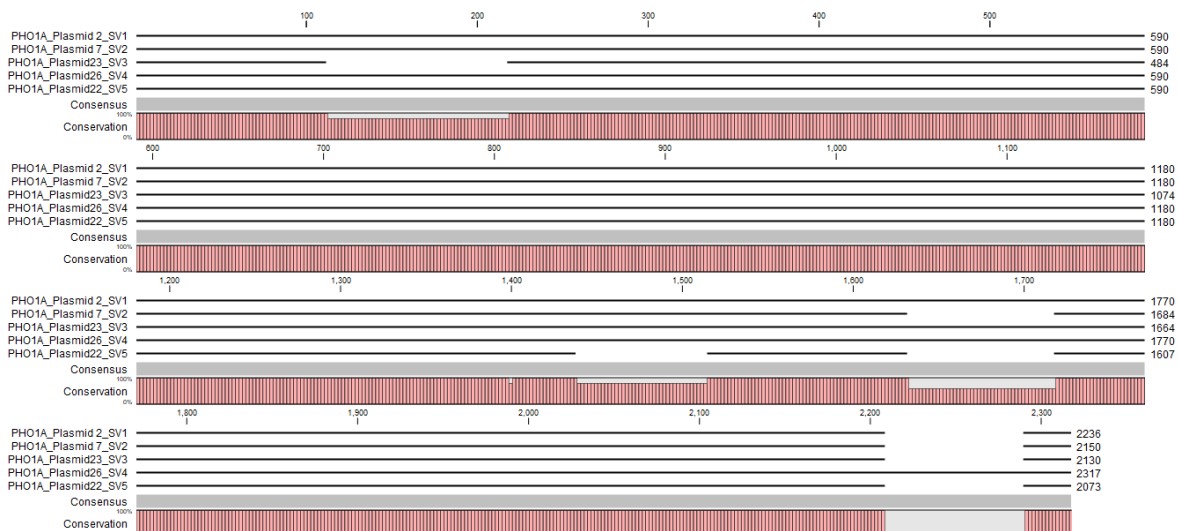

**Figure 6.** Alignment of five *PHO1_A* splice variants in CLC Sequence Viewer 7.8.1 (Qiagen). Row from top: splice variants 1, 2, 3, 4, and 5 for plasmids 2, 7, 23, 26 and 22, respectively.

Conserved domain search in the *PHO1_A* splice variants identified SPX (pfam03105) and EXS (pfam03124) domains in proteins encoded by transcript 2, 23, and 26; however, EXS (pfam03124) is absent in 7 and 22. Both domains were present in *PHO1_B* (transcript 12), whereas *PHO1_D* (transcript 24) truncated protein contained the SPX (pfam03105) domain only.

Three out of four groups of *PHO1* paralogs were expressed in leaf tissue of *B. rapa* under P+ and P− circumstances. Transcript 2, along with four other splice variants, transcript 7, 23, 22, and 26, was designated as *PHO1_A*. *PHO1_A* (XM_009150437) and *PHO1_C* (XM_018652651) are nearly similar sequences except for seven SNPs. However, *PHO1_C* contains an open reading frame of 2025 bp, compared to *PHO1_A*, which is 2445 bp long. As all five transcripts designated as *PHO1_A* ranged in size from 2130 to 2317 without any SNP, it indicates the absence of *PHO1_C* transcripts in the *B. rapa* leaf cDNA library. This result is consistent with previous evidence (Figure 1).

Colony PCR and restriction analysis of 30 colonies with subsequent sequencing of seven chosen plasmids produced full-length *PHO1_A* transcripts along with four splice variants, indicating a function of *PHO1_A* in P regulation. The splice variants may have a role in gene regulation rather than encoding a protein based on the evidence that some of the splice variants do not encode full proteins.

Strong homology of transcript 12 in *B. napus* and *B. oleracea* and inadequate coverage and identity in *B. rapa* genome are unexpected. It may be attributed to the poor efficiency of short-read sequencing in the annotation of duplicated genes with considerable homology. As previously noted, transcript 24 has been assigned to *PHO1_D*. However, the full-length sequence could not be captured. The transcript might not be very prevalent in leaves. Screening of the cDNA library from other tissues, like roots, may confirm this hypothesis.

The final alignment and phylogenetic tree were generated and clearly show the relationship of all related transcripts to the target genes (Figures 7 and 8).

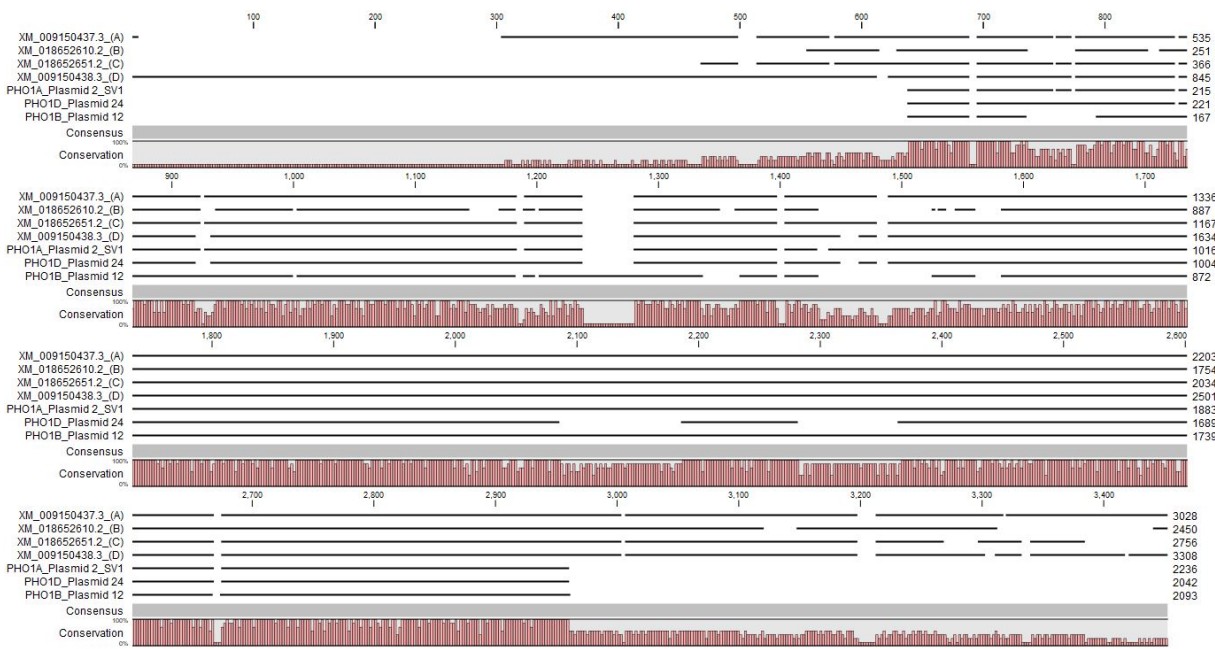

**Figure 7.** Final alignment of *PHO1* paralog genes in CLC Sequence Viewer 7.8.1 (Qiagen). Row from top: XM_018652651.1 (*PHO1_C*), XM_009150437.2 (*PHO1_A*), Plasmid 2, Plasmid 12, and XM_018652610.1 (*PHO1_B*).

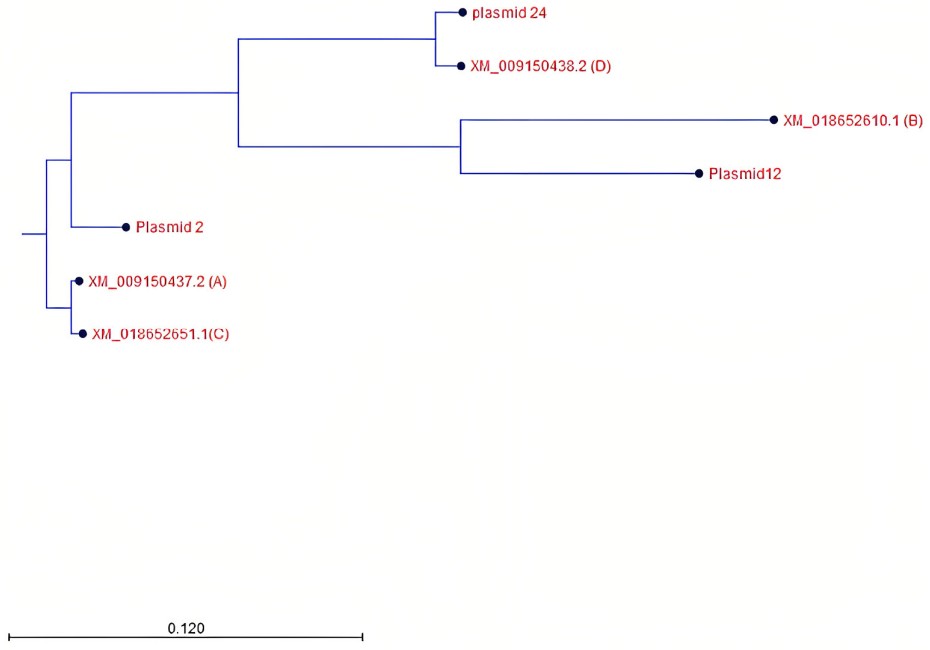

**Figure 8.** Phylogenetic tree of *Brassica rapa* PHO1 plasmids and NCBI paralogs produced from CLC Sequence Viewer 7.8.1 (Qiagen).

## 4. Discussion

Expression quantitative trait loci (eQTL) are genomic areas that affect the variation in gene expression due to sequence polymorphisms in target genes or their regulatory regions [29,30]. Cis-eQTL refers to eQTL that map close to the physical location of their gene of origin. In contrast, trans-eQTL refers to those that map far from their physical position and are frequently found on different chromosomes.

Since a single gene may be linked with one or more eQTL, categorizing eQTL into cis- or trans-eQTL can reveal regulatory networks controlling gene expression. Under

Pi starvation, Pi acquisition, translocation, and remobilization are mediated by the Pi transporter (PHT1 family). In a segregating population, mRNA expression levels were quantified. As a result, eQTL analysis could be used to pinpoint the genomic areas that control the expression level [29]. The eQTL of Pi starvation responses was identified in the *B. rapa* mapping population (IMB 211 × yellow sarson R500), where the progeny was self-crossed for eight generations to develop recombinant inbred lines (RILs) [30].

Analysis of eQTL on *B. rapa* under Pi deficiency has revealed that the number of eQTLs was higher than expected based on physical and genetic maps on chromosomes A06 and A01, where Gene Ontology terms showed high with P metabolism-related genes in A06 and high with chloroplast and photosynthesis-related terms in A01 [25,29].

A list of genes significantly differentially expressed in response to Pi deficiency was generated based on the microarray data of Pi-deficient *B. rapa* R-o-18 [25]. Microsoft Access was used to compare this microarray list of genes to a list of genes in the genomic region surrounding the eQTL hotspot on chromosome A06. This comparison reveals the genes whose expression was influenced by the alteration of Pi availability in this region. Four candidate genes were discovered to be the main regulatory trans-eQTL hotspot and located in tandem. All these genes belong to the *PHO1* gene family. However, short-read transcriptome data revealed only three copies of genes appear (Figure 1). The short-read sequence could be inaccurate due to highly repetitive mapping reads and different types of alignment protocols that could generate the wrong assembly and gene loss [31–34].

The presence of these four paralog copies, *PHO1_A*, *PHO1_B*, *PHO1_C*, and *PHO1_D*, was revealed by de novo assembly of BAC sequences from this region of interest utilizing long-run MinION sequencing technology. Only three transcripts, *PHO1_A*, *PHO1_B*, and *PHO1_D*, were found in *B. rapa* R-o-18 Pi-deficient plants after colony PCR and restriction analysis of 30 colonies, as well as sequencing of seven chosen plasmids.

### 4.1. Potential for Alternatively Spliced PHO1 Transcripts

Based on the results, *PHO1_A* seemed to have five splice variants (Figure 7). The posttranscriptional alterations of *PHO1_A* may have resulted in alternative splicing or structural divergence, which could play a significant role in Pi-gene regulation and contribute to protein diversification. To support these results, additional investigation is required, including the identification of alternatively spliced transcripts and proteins. Furthermore, the occurrence of these four paralogs is the result of gene duplication, which caused the initial identical sequences and functionalities to differ in their regulatory and coding regions [35,36].

Based on the intron–exon structures found in the genes, three processes have been identified that result in alternative splicing or structural divergence. The first is intron–exon gain/loss, which is the process where entire/partial sections of introns/exons are gained by duplicating introns and exons or by reshuffling introns and/or exons. The second is exonization and pseudoexonization, which involve exonic and nonexonic sequences swapping places. The third is intra-exonic insertions or deletions, which may cause the open reading frame and biochemical function to alter [35–37]. A comparison of five paralog transcripts of *PHO1_A* with the genomic sequence found the differences in intron–exon structure occurred due to exonization or pseudoexonization, with the modifications occurring at nucleotide 3494 of the genomic sequence corresponding with the sequence of plasmid 26.

Alternative splicing occurred as a result of Pi deficiency, which is consistent with prior research that implies alternative splicing is necessary for plants to adjust to environmental stress [18]. For instance, in rice, 33% of all rice genes are alternatively spliced, with 58% of these genes having multiple alternative splicing processes, producing a range of transcripts [38].

*4.2. Potential Structural Variation in PHO1 Paralogs*

Analysis of the *B. rapa PHO1* paralog structures showed that the proteins comprise two main domains and share the same features found in all members of the *PHO1* family in *Arabidopsis*. The first half of the protein contains an SPX domain harboring hydrophobic N-terminal, and the second half of the protein contains an EXS domain harboring the hydrophilic C-terminal. Compared to other eukaryotes, PHO1 family members are the only proteins that contain both SPX-EXS domains [17]. SPX-EXS proteins are involved in many biological activities, including Pi uptake, transport, and storage to maintain Pi homeostasis [13,39]. In this study, evidence of different proteins being produced from a single gene was observed.

Analysis of PHO1_A protein sequences using BLASTP identified five possible splice variants that changed the domain structures. Splice variant (sv) 1, sv 2, sv 3, sv 4, and sv 5 encoded proteins of 744, 552, 629, 746, and 490 amino acids, respectively. Three proteins have both SPX-EXS domains, whereas two others (sv 2 and sv 5) contain only an SPX domain (pfam03105) and lack an EXS domain (pfam03124) in the C-terminal. In *Arabidopsis*, four proteins were identified that only contain an SPX domain, namely, SPX1-SPX4 [40]. In rice, six proteins have been identified that only contain an SPX domain (OsSPX1-OsSPX6) [19]. Analysis of PHO1_B showed both SPX and EXS domains were present at both protein terminals, while PHO1_D did not encode a full protein, lacking the EXS domain in the C-terminal. The variety of domain-harboring proteins gives differences in the structure and, therefore, potentially the expression, function, and subcellular localization of these proteins [41].

SPX domains are known to be involved in Pi metabolism and can be found in a variety of evolutionary unrelated proteins [19]. Proteins that only include SPX domains are called SPX proteins. The majority of SPX genes, except AtSPX4 and OsSPX4, are phosphate starvation-induced and are important for signal transduction of inorganic Pi status in plants and strongly upregulated under Pi starvation [38]. SPX functions as a sensor to initiate the activity of the *PHO1* signal to mediate the Pi export to the xylem. Previous studies showed that the SPX domain was not essential for Pi export activity but important in binding Pi to its target sequence [21,42]. In *Arabidopsis*, AtSPX1 and AtSPX2 are localized in the nucleus and connect with PHR1 transcription factor depending on tissue Pi status, thus altering PHR1 binding to its target sequence. Similar results were obtained with the rice ortholog OsPHR2, confirming that the SPX domain may act as a Pi sensor and activate the activity of PHO1 as Pi exporter or Pi signal transduction pathway [43,44].

EXS domains have several functions and are not only important in Pi export activity but have been demonstrated to facilitate long-distance signals from root to shoot to stimulate growth and localize to the Golgi and trans-Golgi network in *Nicotiana benthamiana* [21]. *Pho1* mutant expressing EXS domain showed significant improvement in shoot growth but reduced expression of many genes associated with Pi deficiency [21]. The genes involved included genes encoding enzymes that are responsible for enhancing Pi uptake, Pi recycling, and conservation, including *purple acid phosphatase PAP5* (APase activity), *monogalactosyldiacylglycerol synthase MGD3* (lipid modification), and *SPX1* and *SPX3* (signaling cascade) [21,45–48]. The EXS domain of PHO1 is important for Pi export as well as for playing a role in modulating Pi response through long-distance signaling from root to shoot [19,20,26]. The EXS domain of PHO1 is also crucial for correct localization to the Golgi and trans-Golgi network, although EXS alone could not function to export Pi [21].

**5. Conclusions**

The results obtained in this research can be used to shed new light on the potential evolution of splice variants with altered domain structures within the *B. rapa PHO1* gene family. The data will facilitate future research studying the expression profiles of these variants in different tissue types and environmental situations. This is important since the alternative splicing events vary in their differential regulation between tissues, as well as environmental conditions. Additionally, the clone transcripts contained can be employed in

transgenic research to determine their precise function. The role of SPX and EXS domains can be studied in *PHO1_D* and *PHO1_A* as three splice variants contained both domains, whereas EXS is depleted in the other two splice variants.

**Author Contributions:** Conceptualization, D.S.; methodology, D.S., A.N.S. and R.R.; software, D.S. and A.N.S.; validation, D.S., R.R., A.N.S. and A.B.A.K.; formal analysis, D.S.; investigation, D.S. and A.N.S.; resources, D.S.; data curation, D.S., R.R., A.N.S. and A.B.A.K.; writing—original draft preparation, D.S., R.R. and A.N.S.; writing—review and editing, D.S., A.N.S., R.R. and A.B.A.K.; visualization, R.R., A.N.S. and A.B.A.K.; project administration D.S., R.R. and A.N.S. All authors have read and agreed to the published version of the manuscript.

**Funding:** The research was funded by the Ministry of Higher Education Malaysia (SLAB/715283) and Universiti Sains Malaysia (USM/ASTS).

**Data Availability Statement:** The data presented in this study are openly available in GenBank https://www.ncbi.nlm.nih.gov/nuccore/ON843907 (accessed on 21 May 2023), ON843908, ON843909, ON843910, ON843911, ON843912, ON843913 for PHO1_A_sv1, PHO1_A_sv2, PHO1_A_sv3, PHO1_A_sv4, PHO1_A_sv5, PHO1_B, and PHO1_D, respectively.

**Acknowledgments:** We are thankful to John Hammond and members of the Genetic and Molecular Laboratory, School of Agriculture, Policy and Development, University of Reading.

**Conflicts of Interest:** The authors declare that they have no known competing financial interest or personal relationships that could have appeared in the work reported in this paper.

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
