# Peer review of "Elucidating Quadruplication Event of PHO1 Gene: A Key Regulator of Plant Phosphate Translocation in Brassica rapa"

_horticulturae, doi:10.3390/horticulturae9070845_

Round 1

Reviewer 1 Report

The manuscript is interesting and focuses on the Brassica rapa homolog of the PHO1 gene. I have several comments.

1. Some low-resolution figures in this draft make it difficult to understand the figure details. Please provide high-resolution figures in the revised version.

2. Figure 6, the variants 4, 5, consensus, and conservation of the last row are missing.

3. Figure 6, the alignment between variants 1 and 4 is the same, do they have some nucleotide change?

4. The detailed method to build the phylogenetic tree should be included in the method section.

5. The writing should be improved.

Author Response

Comments and Suggestions for Authors (See attachment file for the revised manuscript with track changes)

The manuscript is interesting and focuses on the Brassica rapa homolog of the PHO1 gene. I have several comments.

  1. Some low-resolution figures in this draft make it difficult to understand the figure details. Please provide high-resolution figures in the revised version.
  • Thank you for the comment. High-resolution figures of Figure 1, Figure 2, and Figure 7 have replaced the previous versions.
  1. Figure 6, the variants 4, 5, consensus, and conservation of the last row are missing.
  • Figure 6: Replaced the previous version with the new version of figure 6, showing the missing consensus and conservation of the last row.
  1. Figure 6, the alignment between variants 1 and 4 is the same, do they have some nucleotide change?
  • Yes, they have some nucleotide changes. There are some differences between variants 1 and 4. The new version of Figure 6 showed differences between variants 1 and 4 at the end of the alignment.
  1. The detailed method to build the phylogenetic tree should be included in the method section.
  • Thank you for the suggestion. The method for phylogenetic tree construction was added in section 2.4.5 (Phylogenetic tree construction).

Comments on the Quality of English Language

  1. The writing should be improved.
  • Thank you for your suggestion. We have submitted the manuscript for proofreading to Enago. The revised version is the proofread and improved version of the manuscript.

proofread

Reviewer 2 Report

The present research investigated PHO1 short read sequencing data of Brassica rapa to find howed four PHO1 paralog genes located in tandem with very high sequence similarity, namely PHO1_A, PHO1_B, PHO1_C, and PHO1_D.  Analysis of long read BAC data verified the quadruplicating tandem locations of PHO1 gene in the genome sequence. Transcript analysis revealed only three groups of PHO1 paralogs (ortholog AT1G14040 in Arabidopsis); PHO1_A, PHO1_B, and PHO1_D expressed in B. rapa leaf tissues under Pi-deficiency. Five splice variants were detected for PHO1_A with transcript ID XM_009150437.2. Truncated proteins encoded by these splice variants indicated the role of PHO1_A in P regulation rather than encoding protein.

The Introduction section contains sufficient information on POX1 genes and their roll in plants, however it lacks explanation on how the present study is filling the gap in current knowladge and what is the aim of this research. The analyses of the POX genes made in several different softwares and presented in Results are not clear and the figures should be exported from softwares. The Authors should explain why they analysed only 30 colonies on PCR and if they have chosen more would that potentialy discover the missing PHO1_C.

The quality of English language is in a need for improvement. 

Author Response

R2_Comments and Suggestions for Authors (See attachment file for the revised manuscript with track changes)

The present research investigated PHO1 short read sequencing data of Brassica rapa to find howed four PHO1 paralog genes located in tandem with very high sequence similarity, namely PHO1_A, PHO1_B, PHO1_C, and PHO1_D.  Analysis of long read BAC data verified the quadruplicating tandem locations of PHO1 gene in the genome sequence. Transcript analysis revealed only three groups of PHO1 paralogs (ortholog AT1G14040 in Arabidopsis); PHO1_A, PHO1_B, and PHO1_D expressed in B. rapa leaf tissues under Pi-deficiency. Five splice variants were detected for PHO1_A with transcript ID XM_009150437.2. Truncated proteins encoded by these splice variants indicated the role of PHO1_A in P regulation rather than encoding protein.

The Introduction section contains sufficient information on POX1 genes and their roll in plants, however it lacks explanation on how the present study is filling the gap in current knowledge and what is the aim  

  • Thank you for your valuable comment. We have added more explanation in the introduction part.

“In this study, we employ the long read sequencing to find the complex structural variants in the genomic regions of Brassica sp. by examining the potential genes at trans-eQTL hotspots. The long reads were processes to elucidate the four copies of PHO1 genes located in tandem in the genome sequence. Furthermore, we defined these genes using transcript amplification and cloning to clarify the transcript presence. It is important to explain the genetic components responsible for phosphate deficiency to further manipulate the genes to improve phosphate use efficiency (PUE) in plants [25,26]."

The analyses of the POX genes made in several different softwares and presented in Results are not clear and the figures should be exported from softwares.

  • Thank you for your suggestion. We have changed Figure 1, 2, 6 and 7 with figures with high resolution versions.

The Authors should explain why they analysed only 30 colonies on PCR and if they have chosen more would that potentialy discover the missing PHO1_C.

  • 30 colonies are statistically acceptable as a representative of the sample. Based on the restriction fingerprints, several variants were found, and the analysis of the transcript can confirm the unavailability of the PHO1_C transcript in the leaf tissues of rapa.

Comments on the Quality of English Language

The quality of English language is in a need for improvement. 

  • Thank you for your suggestion. We have submitted the manuscript for proofreading to Enago. The revised version is the proofread and improved version of the manuscript.
  • proofread

Reviewer 3 Report

In this manuscript, authors described the Elucidating quadruplication event of PHO1 gene; A key regulator of plant phosphate translocation in Brassica rapa. In this study, authors analyze the short read sequencing data of Brassica rapa showed four PHO1 paralog genes located in tandem with very high sequence similarity, namely PHO1_A, PHO1_B, PHO1_C, and PHO1_D. However, only three transcripts are available based on short-read genomic sequence data. Using Oxford Nanopore MinION long read sequencer to sequence five bacterial artificial chromosomes (BAC) enables the improvement of de novo assembly and identification of structural variations. Analysis of long-read data verified the quadruplicating tandem locations of PHO1 gene in the genome sequence. Transcript analysis revealed only three groups of PHO1 paralogs (ortholog AT1G14040 in Arabidopsis); PHO1_A, PHO1_B, and PHO1_D expressed in B. rapa leaf tissues under Pi-deficiency. Five splice variants were detected for PHO1_A with transcript ID XM_009150437.2. Truncated proteins encoded by these splice variants indicated the role of PHO1_A in P regulation rather than encoding a protein.

The manuscript is not suited for publication at this point in time as it is still in the early stage also there is no functional analysis for this study. Cloning is not clear and has dodgy expression results and might not be enough for publication.

Need to improve

Author Response

R3_Comments and Suggestions for Authors (See attachment file for the revised manuscript with track changes)

In this manuscript, authors described the Elucidating quadruplication event of PHO1 gene; A key regulator of plant phosphate translocation in Brassica rapa. In this study, authors analyze the short read sequencing data of Brassica rapa showed four PHO1 paralog genes located in tandem with very high sequence similarity, namely PHO1_APHO1_BPHO1_C, and PHO1_D. However, only three transcripts are available based on short-read genomic sequence data. Using Oxford Nanopore MinION long read sequencer to sequence five bacterial artificial chromosomes (BAC) enables the improvement of de novo assembly and identification of structural variations. Analysis of long-read data verified the quadruplicating tandem locations of PHO1 gene in the genome sequence. Transcript analysis revealed only three groups of PHO1 paralogs (ortholog AT1G14040 in Arabidopsis); PHO1_APHO1_B, and PHO1_D expressed in B. rapa leaf tissues under Pi-deficiency. Five splice variants were detected for PHO1_A with transcript ID XM_009150437.2. Truncated proteins encoded by these splice variants indicated the role of PHO1_A in P regulation rather than encoding a protein.

The manuscript is not suited for publication at this point in time as it is still in the early stage also there is no functional analysis for this study.

Cloning is not clear and has dodgy expression results and might not be enough for publication.

  • Thank you for your comment. The cloning of the PHO1 transcript followed the routine cloning work as follows.

Identification and characterization of PHO1 homolog transcripts

Total RNA from leaf tissues of B. rapa was transcribed into cDNA. A gradient PCR was conducted with both primer sets at 60 and 65 ºC annealing temperature using cDNA from P+ and P- treatments for optimization. Both primer sets amplified multiple bands in expected size ranges at 60ºC (Fig. 1). However, primer set PHO_1_F and PHO_2224_R produced brighter and distinct bands. Fig. 1 A). A band of unexpected size of around 4 Kb was also amplified that might have originated from alternative splicing of PHO1 or could be an entirely non-related transcript. 

1

Figure 1. Electrophoresis gel image of PHO1 Polymerase Chain Reaction (PCR) of cDNA from Brassica rapa R-o-18 leaf samples. A. Lane 1 from left: FastRuler Middle Range DNA Ladder (L), lane 2 to 9 is PCR for cloning PHO1 transcripts. B. Gel of an aliquot of the PCR products after optimizing PCR targeting PHO1_F+ PHO_2224_R.

Cloning PHO1 transcripts

Two optimized PCR reactions of P+ and P- cDNA and PHO1_F+ PHO_2224_R verified the sizes found in earlier PCR (Fig. 1). Both PCR products were pooled to represent both treatments, purified, and ligated into the pCR4 BLUNT TOPO vector. Topo10 E. coli cells were transformed with the ligation mixture and showed good growth (Fig. 2).

Figure 2. Agar plates; A. LB agar with 50 mgL-1 of Kanamycin plate containing 20 µL transformed E. coli culture. B. LB agar with 50 mgL-1 of Kanamycin plate containing 200 µL transformed E. coli culture.

Colony PCR and restriction analysis

Plasmid DNA was isolated from ten clones based on colony PCR analysis. Restriction maps of the targeted region of XM_009150437 (PHO1_A), XM_018652610 (PHO1_B), XM_018652651 (PHO1_C) and XM_009150438 (PHO1_D) in pCR4 BLUNT TOPO vector were constructed using Vector NTI software (Fig. 3). Restriction of 1 µg plasmid DNA of the ten selected clones confirmed the findings of the colony PCR, hence these were sent for sequencing.

Table. Transcript length expected cut sites and fragment length of PHO1 homologs restricted with HindIII enzymes.

Gene

Transcript length (bp)

Cut sites

Fragment Size (bp)

PHO1_A

2224

50, 970

50, 920, 1254

PHO1_B

2044

80, 1593, 1677

80, 84 367, 1513

PHO1_C

2152

46, 966

46, 920, 1186

PHO1_D

2227

50, 479, 938, 961, 1772

23, 50, 429, 455, 459, 805

Figure 3. Restriction maps of the targeted region of A. XM_009150437 (PHO1_A) and XM_018652610 (PHO1_B), B. XM_018652651 (PHO1_C) and XM_009150438 (PHO1_D) = pCR4 BLUNT TOPO sequencing vector.

  • We have discussed the functional analysis of PHO1 protein in the discussion part although not in-depth, as the main aim of this research is to elucidate the structural variations of PHO1 genes as a major regulator at eQTL-hotspots (see: Introduction-last paragraph).

Comments on the Quality of English Language

Need to improve

Thank you for your suggestion. We have submitted the manuscript for proofreading to Enago. The revised version is the proofread and improved version of the manuscript

Reviewer 4 Report

I reviewed the manuscript titled "Elucidating quadruplication event of PHO1 gene; A key regulator of plant phosphate translocation in Brassica rapa" and found interesting however there are a lot of concerns/errors inside the manuscript preventing the manuscript to be publish in the current form. The major concern include 

1. Manuscript is full of flaws/errors without any concept while the data is mosly bioinformatics lacking experimental evidences. 

2. Why there are so many figures of irrelevent things like Multiple sequence alignment. 

3. The phylogentic tree is confusing. 

4. Why to add the sequencer name in abstract as sequencing is very common now a days.

5. What is the meaning of this sentences given in the abstract "Truncated proteins encoded by these splice variants indicated the role of PHO1_A in P regulation rather than encoding protein."

6. Which ladders have been used and why the ladders are given with a single/few band in gel images?

Quality of english language is low.

Author Response

R4_Comments and Suggestions for Authors (See attachment file for the revised manuscript with track changes)

I reviewed the manuscript titled "Elucidating quadruplication event of PHO1 gene; A key regulator of plant phosphate translocation in Brassica rapa" and found interesting however there are a lot of concerns/errors inside the manuscript preventing the manuscript to be publish in the current form. The major concern include 

  1. Manuscript is full of flaws/errors without any concept while the data is mosly bioinformatics lacking experimental evidences. 
  • Thank you for your comment. In this study, we employ the long read sequencing to find the complex structural variants in the genomic regions of Brassica sp. by examining the potential genes at trans-eQTL hotspots. The long reads were processes to elucidate the four copies of PHO1 genes located in tandem in the genome sequence. Furthermore, we defined these genes using transcript amplification and cloning to clarify the transcript presence. It is important to explain the genetic components responsible for phosphate deficiency to further manipulate the genes to improve phosphate use efficiency (PUE) in plants. (See: Introduction_last paragraph)
  1. Why there are so many figures of irrelevent things like Multiple sequence alignment. 
  • The alignment figures are present in the manuscript to show the similarities/differences of multiple sequences. For example, in Figure 2, the alignments showed the high similarities of four candidate genes found in Brassica genome from the results of short read sequencing. Therefore, this study was conducted and used long read sequencing method to improve the de novo assembly or detect errors due to structural variants, duplication errors, or pseudogenes. In Figure 6, multiple alignment shows the differences between five splice variance sequences found for PHO1_A. While in Figure 7, the multiple alignments showed the differences between all PHO1 paralog genes investigated in this research.
  1. The phylogenetic tree is confusing. 
  • Thank you for your comment. The phylogenetic tree was constructed using neighbor joining method.

“All sequences were aligned, and the neighbor-joining method was used to perform cluster analysis. The phylogenetic tree was created based on the distance data using CLC Sequence Viewer 7.8.1 (Qiagen) software.” (See section 2.4.5 Phylogenetic tree construction).

  1. Why to add the sequencer name in abstract as sequencing is very common now a days.
  • Thank you for your comment. We have removed the sequencer name in the abstract
  1. What is the meaning of this sentences given in the abstract "Truncated proteins encoded by these splice variants indicated the role of PHO1_A in P regulation rather than encoding protein."
  • Thank you for your query. We have amended the sentence to “These splice variants’ truncated proteins demonstrated PHO1_A’s function in P control as opposed to protein encoding.”
  1. Which ladders have been used and why the ladders are given with a single/few band in gel images?
  • The ladder used was Thermo Scientific Fast Ruler Middle Range DNA Ladder. The ladder consists of five blunt-end individual DNA fragments; 5000, 20000, 850, 400 and 100 base pairs

Comments on the Quality of English Language

Quality of english language is low.

  • Thank you for your suggestion. We have submitted the manuscript for proofreading to Enago. The revised version is the proofread and improved version of the manuscript.

Round 2

Reviewer 1 Report

The revision has answered my questions. I suggest for publication. Thanks for the authors‘ efforts.

Author Response

Thank you and we appreciate your insightful comments. 

Reviewer 2 Report

The manuscript has been revised according to the suggestions and it is know ready to be accepted. 

Moderate revisions of English language are required.

Author Response

Thank you and we really appreciate your insightful comments.

We have submitted the manuscript to Enago and subsequently to Grammarly Premium for proofread and revision of the English language.

Reviewer 3 Report

Still very low-quality data with only bioinformatics not backed by wet lab experiments. Bands are not the same size in colony PCR and cDNA synthesis showed two bands that are not good for cloning. Data is not reliable. Hence not recommended for publication.

Still very low-quality data with only bioinformatics not backed by wet lab experiments. Bands are not the same size in colony PCR and cDNA synthesis showed two bands that are not good for cloning. Data is not reliable. Hence not recommended for publication.

Author Response

Reviewer: Still very low-quality data with only bioinformatics not backed by wet lab experiments. Bands are not the same size in colony PCR and cDNA synthesis showed two bands that are not good for cloning. Data is not reliable. Hence not recommended for publication.

Answer: Thank you for your comments. Bands have different sizes in colony PCR showed the several variants of transcript available.

cDNA synthesis showed two bands.

Answer: We used the right primer so that we choose band of intereast by the sequence length, and cut the band that has correct sequence length for cloning using TOPO kit.

We have revised the manuscript and hope you can consider it for publication.

Reviewer 4 Report

The manuscript has been improved enough and I would like to recommend its publication in its current form. 

English language is fine. 

Author Response

(The authors gave the same response as above.)
